# SigLIP-HD by Fine-to-Coarse Supervision

**Lihe Yang**[1]     **Zhen Zhao**[2][†]     **Hengshuang Zhao**[1][†]
[1]The University of Hong Kong     [2]Shanghai AI Laboratory
https://github.com/LiheYoung/SigLIP-HD

## ABSTRACT

High-quality visual representation is a long-standing pursuit in computer vision. In the context of multimodal LLMs (MLLMs), feeding higher-resolution images can produce more fine-grained visual tokens. However, it introduces additional computational and design complexity, due to multiple forward passes and post-processing of increased tokens. Before simply adopting a higher resolution, have we truly unlocked the model's full perception capability at a *standard* resolution? Therefore, we study an interesting problem: how to achieve *fine* visual perception under lower cost *without larger images*. We present `SigLIP-HD` in this work. The core is a highly simple fine-to-coarse supervision design. We enforce the coarse feature of a mid-resolution image to mimic the fine-grained feature of its high-resolution version. We build this framework on the advanced SigLIP 2 model. Our final model produces better visual tokens at exactly the same inference budget. It is validated on extensive MLLM benchmarks and consistently delivers stronger results than our baseline model, especially on OCR-related tasks.

## 1 INTRODUCTION

From supervised pre-training (Deng et al., 2009), to vision-centric self-supervised learning (Wu et al., 2018; He et al., 2022), to vision-language contrastive learning (Radford et al., 2021), further to hybrid training paradigms (Maninis et al., 2025), the computer vision community keeps pursuing more transferable visual representations. High-quality image embeddings (Tschannen et al., 2025; Siméoni et al., 2025) have fundamentally advanced the development of a wide range of perception and generation tasks (Lin et al., 2014; Yu et al., 2025). In recent years, witnessing the power of LLMs (Achiam et al., 2023), finding better visual tokens for multimodal LLMs (MLLMs) (Liu et al., 2023) is receiving growing attention (Tong et al., 2024b;a). The quality of these tokens is critical for MLLMs to accurately perceive and reason over visual signals.

There are three mainstream approaches to improving the visual representations in MLLMs. The first is to directly pre-train a better vision encoder from scratch with better algorithms (Oquab et al., 2024; Tschannen et al., 2025), more data (Fan et al., 2025; Bolya et al., 2025), or larger models (Siméoni et al., 2025). This line of work requires tremendous resources (*e.g.*, million GPU hours, billion data), which are unaffordable for most researchers. The second approach is leveraging multiple well-trained encoders (Tong et al., 2024b;a; Shi et al., 2025b). Different encoders have their own strengths. CLIPs (Radford et al., 2021) are good at modeling vision-language correspondence, while vision-only models (Oquab et al., 2024) excel at capturing detailed visual correlations. Incorporating them may amplify their distinct advantages while suppressing their drawbacks. Despite being intuitively promising, the actual gain is limited or even negative (Tong et al., 2024a).

The last approach is simply to forward higher-resolution input (Liu et al., 2024a;b; Li et al., 2024). Higher-resolution images can yield more fine-grained visual tokens, making MLLMs see more clearly. This principle is gradually strengthened by state-of-the-art MLLMs (Liu et al., 2025; Wu et al., 2024b; Deitke et al., 2025), from resizing to a fixed larger size (Liu et al., 2024a), to fully preserving the native resolution (Li et al., 2025a; Bai et al., 2025), steadily contributing to better OCR capability. Our work is inspired by this observation, but with fundamentally different roadmaps. We highlight that, although increased image size improves visual perception, it brings significant extra complexity in both compute and design. The image has to be sliced into small tiles (Liu et al., 2024b;

---

[†]Corresponding author

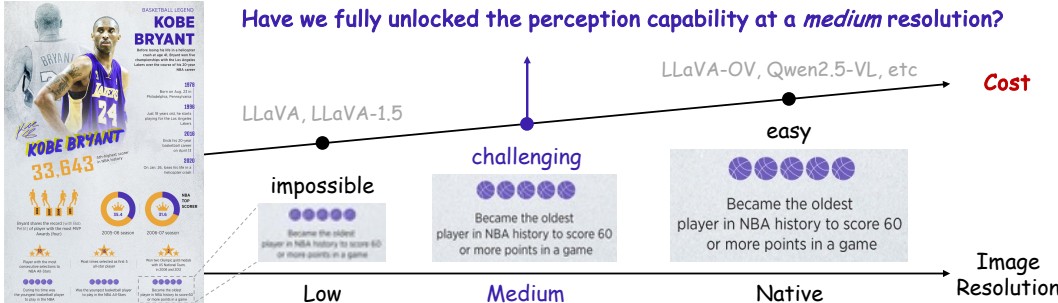

Figure 1: Early MLLMs (Liu et al., 2023; 2024a) resize images to a fixed low resolution (*e.g.*, $336^2$px), while recent works (Li et al., 2025a; Bai et al., 2025; Liu et al., 2025) operate on native resolution with huge costs. But indeed, at a *medium* resolution (*e.g.*, 512px), humans can already understand the content. How to make AI systems achieve this?

Li et al., 2025a) to match the pre-trained vision resolution. This not only requires time-consuming multiple forward passes, but also incurs more visual tokens. To alleviate the LLM's burden, further post-processing, *e.g.*, resampler (Alayrac et al., 2022) and pixel unshuffle (Chen et al., 2024b), is necessary for token compression, making the framework even more complicated.

We reflect on the scaling trend. Before increasing the input resolution, have we fully unlocked the model's perception capability at a *standard (medium)* resolution (*e.g.*, $512^2$px)? Can we achieve *fine-grained* visual perception under lower cost *without larger images*?

At least our human visual system can. As shown in Figure 1, when downsampling the image from its native resolution (1722px in height) to a medium size (512px), although the characters are blurred, we can still accurately understand the content. Therefore, if an AI visual system is developed capable enough to match human performance, it should be able to perceive at such a medium resolution, without requiring a larger image. It will avoid unnecessary computation at high resolutions, reducing token redundancy and improving system efficiency.

So the remaining problem is, how to make vision models acquire such promising perception capability. Our solution is highly simple yet effective. We design a fine-to-coarse supervision mechanism to transfer high-quality visual representation at a high resolution to coarse visual tokens at a standard resolution. Concretely, given a pre-trained vision encoder (Tschannen et al., 2025), we further fine-tune it at a standard resolution by enforcing its output features to mimic the fine-grained features of the corresponding high-resolution images. Through this mechanism, the encoder gradually bootstraps its representations from coarse level to fine level. Our method does not rely on any human or synthetic labels for fine-grained perception (Chen et al., 2025; Choi et al., 2025; Shi et al., 2025a), purely exploring cheap raw images for self-improvement. Without using any auxiliary upsamplers (Fu et al., 2024; Huang et al., 2025), our final model inherits the same structure and IO as our pre-trained model, making it very easy and efficient to deploy in complex systems. Users only need to modify the checkpoint path to ours, and then enjoy better perception capability.

Our main contributions are summarized below:

- We reflect on the scaling trend of image resolution in MLLMs. Beyond using high-resolution images for high-quality visual tokens, we study an intriguing problem: how to achieve fine-grained perception for MLLMs under lower cost *without larger images*.
- We present a highly simple fine-to-coarse supervision mechanism to strengthen perception capability at a standard resolution by imitating "teacher" tokens of high-resolution images.
- We build our method on the competitive SigLIP 2 encoder (Tschannen et al., 2025). With the same inference cost, our SigLIP-HD delivers better results across diverse MLLM benchmarks under various protocols, especially for scenarios that favor fine-grained perception.

Following the philosophy of our fine-to-coarse supervision, in Section 2, we first investigate *what* good fine-grained representation to learn. Then in Section 3, we present our framework on *how* to learn from the fine-grained representation.

Table 1: Comparison between CLIP (Radford et al., 2021) and SigLIP 2 (Tschannen et al., 2025) on MLLM benchmarks following the training protocol of LLaVA-1.5-7B (Liu et al., 2024a). We evaluate their top-performed versions. *We do not include OCR tokens into prompts, following (Zhang et al., 2025). We divide the $MME^P$ score by 20 before averaging.

| | OCR & Chart | | | | | General | | | | | Knowledge | | | |
|---|---|---|---|---|---|---|---|---|---|---|---|---|---|---|
| **Vision encoder** | DocVQA | ChartQA | TextVQA* | InfoVQA | TextCaps | HRBench | RWQA | GQA | $MME^P$ | POPE | MMBench | $SQA^I$ | AI2D | **Avg** |
| CLIP-L/14-336px | 22.4 | 17.5 | 46.8 | 20.8 | 69.0 | 36.9 | 56.0 | 63.0 | 1520.9 | 86.9 | 66.2 | 70.2 | 55.9 | 52.9 |
| SigLIP 2-L/16-384px | 25.1 | 16.4 | 55.1 | 21.5 | 70.6 | 39.3 | 56.0 | 63.9 | 1538.7 | 87.2 | 67.5 | 69.1 | 58.7 | 54.4 |
| SigLIP 2-400m/14-384px | 29.5 | 19.0 | 59.7 | **22.9** | **71.5** | 42.2 | 57.4 | **64.8** | 1538.4 | 87.8 | 67.8 | 70.8 | 58.2 | **56.0** |
| SigLIP 2-400m/16-512px | **32.2** | **19.3** | **61.0** | 22.1 | 71.1 | 41.3 | 57.5 | 62.8 | 1529.2 | **87.8** | 67.7 | 69.9 | 56.0 | 55.8 |
| SigLIP 2-G/16-384px | 26.7 | 18.5 | 58.2 | 21.4 | 71.3 | **42.7** | **59.9** | 64.6 | **1559.4** | 87.0 | **68.6** | **71.1** | **59.9** | 56.0 |

Table 2: Comparison among different image scales to obtain final features based on SigLIP 2-512px. Here, we interpolate high-resolution features to base scale and average multi-scale features (if any) to ensure the same number of visual tokens across all settings.

| | OCR & Chart | | | | | General | | | | | Knowledge | | | |
|---|---|---|---|---|---|---|---|---|---|---|---|---|---|---|
| **Image scale** | DocVQA | ChartQA | TextVQA | InfoVQA | TextCaps | HRBench | RWQA | GQA | $MME^P$ | POPE | MMBench | $SQA^I$ | AI2D | **Avg** |
| $512^2$ | 32.2 | 19.3 | 61.0 | 22.1 | 71.1 | 41.3 | 57.5 | 62.8 | 1529.2 | 87.8 | **67.7** | 69.9 | 56.0 | 55.8 |
| $512^2+1024^2$ | **37.0** | **21.2** | 64.2 | **23.8** | 71.0 | 47.9 | **59.6** | 64.9 | 1560.1 | 88.0 | 67.4 | 71.0 | **56.2** | **57.7** |
| $512^2+1024^2+1536^2$ | 36.1 | 19.2 | **64.4** | 23.1 | 71.0 | 48.4 | 59.4 | 64.8 | **1565.1** | **88.7** | 67.2 | **71.4** | 56.2 | 57.6 |
| $512^2+1024^2+1536^2+2048^2$ | 35.5 | 20.9 | 64.3 | 23.4 | **71.6** | 46.8 | 59.5 | **65.5** | 1560.0 | 88.0 | 67.0 | 69.8 | 56.1 | 57.4 |
| $1024^2$ | 35.4 | 19.4 | 60.5 | 23.6 | 70.4 | 46.0 | 57.7 | 64.4 | 1500.8 | 88.4 | 66.7 | 69.3 | 55.7 | 56.3 |
| $1536^2$ | 34.4 | 17.7 | 59.1 | 23.4 | 69.5 | **49.3** | 59.1 | 64.6 | 1502.7 | 88.3 | 65.8 | 69.0 | 54.9 | 56.2 |

## 2 WHAT IS GOOD REPRESENTATION TO LEARN?

In this work, we build our fine-to-coarse supervision methodology on the SigLIP 2 encoder (Tschannen et al., 2025), or more specifically, its So400m/16-512px version. This model of 429M parameters takes a $512^2$px image as its input, and produces $32^2$ visual tokens under patch size 16. As compared in Table 1, its So400m/16-512px version is much better than other counterparts in important OCR scenarios (Mathew et al., 2021; Singh et al., 2019), due to its larger resolution. So we mainly use this version for validation. Furthermore, to be more convincing, we also provide our results built on the legacy model OpenAI-CLIP (Radford et al., 2021) (Table 11).

Prior to fine-to-coarse supervision, a primary step is to find out what optimal fine-grained features to learn. Therefore, this section presents necessary pilot investigations.

**Which image scales should be used to obtain fine-grained features?** Common practices (Liu et al., 2024b; Chen et al., 2024b) suggest that it is beneficial to incorporate fine-grained but local features from high-resolution images with coarse but global features from base-scale images (*i.e.*, thumbnails). We feed pre-trained SigLIP 2-512px with a single-scale image or multi-scale images: 1) $512^2$ scale, 2) $512^2+1024^2$ scales, 3) $512^2+1024^2+1536^2$ scales, 4) $512^2+1024^2+1536^2+2048^2$ scales, 5) $1024^2$ scale, and 6) $1536^2$ scale. By default, for high-resolution images, we split them into non-overlapping base-scale images for inference. Then, we interpolate the re-assembled high-resolution features to the same shape as base-scale features ($32^2$ tokens). Finally, we average multi-scale features (if any) to ensure the same number of visual tokens across all settings.

As shown in Table 2, even with the same number of tokens, introducing high-resolution images (*e.g.*, $512^2 \rightarrow 512^2+1024^2$) facilitates most benchmarks (10 out of 12), especially for OCR and chart tasks. But such improvement saturates or disappears when further adding the $4\times$ ($2048^2$) scale, showcasing it is suboptimal to blindly scale up the resolution. The single high-resolution view is generally inferior

Table 3: Comparison among different inference strategies for high-resolution images ($1024^2$). PE interpolation: interpolate the pre-trained positional embeddings to the targeted resolution.

| Infer high-res image | OCR & Chart | | | | | General | | | | | | Knowledge | | Avg |
|---|---|---|---|---|---|---|---|---|---|---|---|---|---|---|
| | DocVQA | ChartQA | TextVQA | InfoVQA | TextCaps | HRBench | RWQA | GQA | $MME^P$ | POPE | MMBench | $SQA^I$ | AI2D | |
| Sliding (w/o overlap) | **37.0** | 21.2 | **64.2** | **23.8** | 71.0 | **47.9** | **59.6** | 64.9 | **1560.1** | 88.0 | 67.4 | **71.0** | 56.2 | **57.7** |
| Sliding (w/ half overlap) | 34.8 | **21.4** | 63.2 | 23.3 | 71.3 | 43.9 | 56.2 | **65.0** | 1535.3 | **88.2** | **68.4** | 70.3 | 56.4 | 56.9 |
| Entire (PE interpolation) | 33.5 | 21.2 | 63.2 | 22.7 | **71.8** | 41.5 | 58.6 | 64.6 | 1548.2 | 87.9 | 67.2 | 69.8 | **57.3** | 56.7 |

Table 4: Comparison among different fusion strategies for multi-scale features from $512^2 + 1024^2$ scales. We first employ interpolation or pixel unshuffling (Shi et al., 2016) to downsample high-resolution features, and then average or concatenate (channel-wise) them with base-scale features.

| Fuse multi-scale features | OCR & Chart | | | | | General | | | | | | Knowledge | | Avg |
|---|---|---|---|---|---|---|---|---|---|---|---|---|---|---|
| | DocVQA | ChartQA | TextVQA | InfoVQA | TextCaps | HRBench | RWQA | GQA | $MME^P$ | POPE | MMBench | $SQA^I$ | AI2D | |
| Interpolate + average | **37.0** | **21.2** | **64.2** | **23.8** | 71.0 | **47.9** | **59.6** | **64.9** | **1560.1** | 88.0 | 67.4 | **71.0** | 56.2 | **57.7** |
| Interpolate + concat | 34.8 | 21.2 | 63.2 | 22.9 | **71.3** | 44.3 | 57.1 | 64.5 | 1555.5 | **88.2** | **68.3** | 70.9 | **57.3** | 57.1 |
| Pixel unshuffle + concat | 35.9 | 20.4 | 63.8 | 22.9 | 70.9 | 42.1 | 59.5 | 64.5 | 1542.4 | 88.1 | 66.5 | 69.9 | 56.4 | 56.8 |

to multi-scale views, highlighting the necessity of maintaining a global view. Lastly, we can observe some benchmarks, *e.g.*, MMBench (Liu et al., 2024c), even do not prefer fine-grained features.

**How to infer a model on high-resolution images?** To apply a model trained on base scale (*e.g.*, 512px) to higher-resolution input, existing works (Liu et al., 2024b; Chen et al., 2024b) typically slide a base-scale window across the entire image with a non-overlapping stride. However, alternative strategies exist. For example, following practices in dense prediction tasks (Cordts et al., 2016), using overlapping sliding windows can enhance local feature coherence and reduce boundary artifacts. Besides, we can directly interpolate the pre-trained positional embeddings to match the high-resolution input (Bai et al., 2023), though this may introduce distortion in fine-grained spatial relationships.

We compare these inference approaches in Table 3. As expected, interpolating positional embeddings underperforms the basic sliding window strategy. Surprisingly, overlapping windows also degrade performance, contrary to lessons from dense prediction tasks. While the exact cause remains unclear, we hypothesize two potential factors: 1) inconsistent token distributions: overlapping regions receive ensemble treatment while others do not, or 2) positional embedding conflicts for tokens appearing in multiple windows. Finally, the best practice is still using non-overlapping sliding window.

**How to fuse features from multiple resolutions?** For multi-scale features, some works (Shi et al., 2024; Liu et al., 2025) first interpolate high-resolution features to the same scale as base-resolution features and then concatenate them along the channel dimension. Other than concatenation, we can simply average them. Besides, pixel unshuffle (Shi et al., 2016) is another choice (Chen et al., 2024a) to downsample features.

As compared in Table 4, the simplest "interpolate + average" practice stands out as the best. Beyond delivering better performance, this strategy produces ensembled features with the same dimension as base-scale features, eliminating the need for a projection head (Ranzinger et al., 2024) in later learning. Indeed, there are more sophisticated fusion options, *e.g.*, using weighted average instead of a simple mean. We leave these investigations to our final main experiments (Table 10).

**Summary.** Our pilot studies demonstrate that the optimal multi-scale configuration is two or three scales ($512^2 + 1024^2$ or $512^2 + 1024^2 + 1536^2$). For high-resolution inference, non-overlapping sliding window yields the best results. Finally, high-resolution features should be interpolated and then averaged with the base-scale features to produce the final features. Based on these findings, we will next present our fine-to-coarse supervision mechanism.

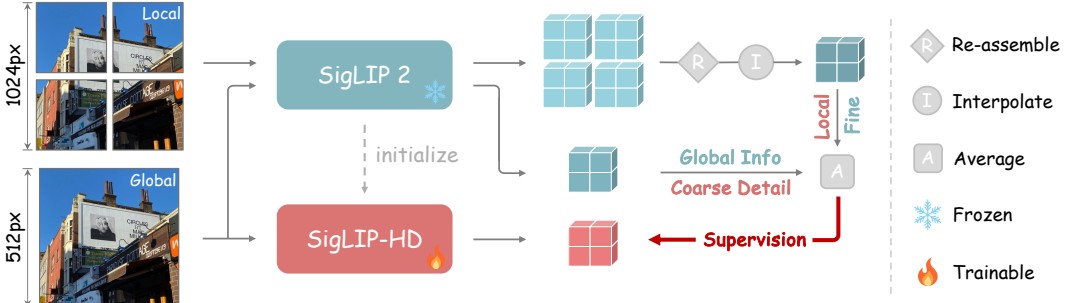

Figure 2: Overview of our fine-to-coarse supervision framework for training our SigLIP-HD. The frozen pre-trained SigLIP 2 encoder is inferred on multi-scale images to produce high-quality fine-grained features. Our SigLIP-HD is trained to mimic the features at a standard resolution ($512^2$).

## 3   SIGLIP-HD

Our core methodology is fine-to-coarse supervision. As illustrated in Figure 2, our framework is very simple. It comprises two branches, one inference branch to produce high-quality visual features, and one trainable branch to learn towards the better features.

**Inference branch.** As studied in our pilot experiments (Table 2), multi-scale input is better than single-scale input. It can harvest both advantages of high-resolution images (fine-grained details) and base-resolution images (global information), while suppressing their weaknesses (local information and coarse details). Therefore, we feed our pre-trained SigLIP 2-So400m/16-512px encoder with two scales: base scale ($512^2$) and $2\times$ scale ($1024^2$). We do not introduce more scales for inference, because we empirically find that more scales fail to consistently yield stronger results (see Table 9). In addition, too many scales significantly increase the computational burden, *e.g.*, requiring nine more feedforward passes if adding a $3\times$ scale.

The pre-trained SigLIP 2 model is frozen and it is inferred on the multi-scale input images by non-overlapping sliding window if needed (supported by Table 3). We obtain a base-resolution feature $F^b \in \mathbb{R}^{C \times H \times W}$ and a high-resolution feature $F^h \in \mathbb{R}^{C \times 2H \times 2W}$ from the two scales, respectively. To fuse the feature maps of different scales, we first downsample $F^h$ to the same size as $F^b$ via bilinear interpolation. We then produce the final high-quality feature $F^t$ by directly averaging $F^b$ and the downsampled $F^h$. This simple strategy is more effective than other counterparts, such as pixel unshuffle and concatenation, as validated in Table 4.

**Trainable branch.** Our SigLIP-HD in the trainable branch is initialized with the pre-trained SigLIP 2 parameters. It shares exactly the same architecture and input/output as the pre-trained model, without adding any projection modules (Ranzinger et al., 2024). It takes a base-scale image of $512^2$ pixels as input and produces $32^2$ tokens. We denote its feature map as $F^s$. The optimization target of this branch is enforcing $F^s$ to align with the better feature $F^t$ at the patch level.

There are various options in the loss function for feature alignment, such as cosine similarity loss (Fang et al., 2023), or a combination of smooth L1 and cosine similarity loss (Ranzinger et al., 2024). In practice, we observe the strictest and simplest L1 loss performs the best, as compared in Table 8.

**Application scope.** Although our method can polish the visual representation at a standard resolution, we do not anticipate it will entirely bypass the native-resolution practice in MLLMs. Some visual details may be completely lost after down-sampling. Fortunately, our method is fully compatible with existing practices, regardless of using down-sampled resolution (Table 5) or operating on native resolution (Table 6), as they both rely on a pre-trained vision encoder.

## 4   EXPERIMENT

**Implementation details.** We train our fine-to-coarse supervision framework on the 4.5M raw images from the Cambrian-1 collected data (Tong et al., 2024a), since they cover diverse scenarios, including natural images, scene text images, documents, *etc.* We train our SigLIP-HD with an AdamW

Table 5: Comparison between our SigLIP-HD and the most capable SigLIP 2-So400m/16-512px encoder (Tschannen et al., 2025). We follow the two-stage training pipeline of LLaVA (Liu et al., 2024a) with Vicuna-1.5-7B (Zheng et al., 2023). We try to freeze (✓) or unfreeze (✗) the vision encoder to provide better insights into our advantages.

| SFT data | Freeze | Encoder | OCR & Chart | | | | | General | | | | | | Knowledge | | Avg |
| | | | DocVQA | ChartQA | TextVQA | InfoVQA | TextCaps | HRBench | RWQA | GQA | MME$^P$ | POPE | MMBench | SQA$^I$ | AI2D | |
| LLaVA-1.5 | ✓ | SigLIP 2 | 32.2 | 19.3 | 61.0 | 22.1 | 71.1 | 41.3 | 57.5 | 62.8 | 1529.2 | 87.8 | 67.7 | 69.9 | 56.0 | 55.8 |
| | | SigLIP-HD | 34.7 | 20.2 | 63.1 | 23.0 | 71.6 | 46.2 | 59.5 | 64.3 | 1554.0 | 88.1 | 67.7 | 71.5 | 58.0 | 57.4 |
| LLaVA-1.5 | ✗ | SigLIP 2 | 32.8 | 20.8 | 62.3 | 22.1 | 71.9 | 42.9 | 56.5 | 63.4 | 1479.0 | 88.3 | 67.6 | 70.6 | 56.3 | 56.1 |
| | | SigLIP-HD | 35.2 | 21.8 | 63.9 | 23.2 | 72.0 | 43.9 | 58.8 | 64.1 | 1550.1 | 88.8 | 67.0 | 70.4 | 57.3 | 57.2 |
| LLaVA-NeXT | ✓ | SigLIP 2 | 54.2 | 58.2 | 64.7 | 23.5 | 67.2 | 43.3 | 58.0 | 63.4 | 1490.8 | 87.8 | 68.2 | 70.2 | 68.3 | 61.7 |
| | | SigLIP-HD | 55.2 | 60.9 | 64.9 | 25.4 | 67.6 | 46.2 | 59.8 | 64.4 | 1536.7 | 88.1 | 68.6 | 72.6 | 68.7 | 63.0 |
| LLaVA-NeXT | ✗ | SigLIP 2 | 56.0 | 61.6 | 65.8 | 23.2 | 67.5 | 43.5 | 60.8 | 63.8 | 1532.1 | 88.1 | 68.4 | 71.8 | 69.2 | 62.8 |
| | | SigLIP-HD | 59.6 | 65.2 | 65.7 | 25.5 | 68.1 | 48.3 | 59.2 | 64.4 | 1558.3 | 88.6 | 70.3 | 74.4 | 70.5 | 64.4 |

Table 6: Evaluation under the **AnyRes** (Li et al., 2025a) strategy (*i.e.*, operating on native resolution). Here we use LLaVA-NeXT data for visual instruction tuning and unfreeze the vision encoder.

| Encoder | DocVQA | ChartQA | TextVQA | InfoVQA | TextCaps | GQA | MME$^P$ | POPE | SQA$^I$ | AI2D | Avg |
|---|---|---|---|---|---|---|---|---|---|---|---|
| SigLIP 2 | 67.6 | 63.9 | 66.9 | 27.2 | 65.6 | 61.1 | 1431.6 | 87.4 | 70.7 | 65.8 | 64.8 |
| SigLIP-HD | 69.7 | 67.4 | 68.4 | 27.7 | 65.8 | 61.9 | 1452.6 | 87.9 | 72.3 | 69.3 | 66.3 |

optimizer (Loshchilov & Hutter, 2019), with an initial learning rate of 5e-5 and weight decay of 0.04. The model is trained for 90K iterations with a total batch size of 512. We use the cosine learning rate scheduler with a warm-up period of 4K iterations. We exactly follow the image pre-processing pipeline of SigLIP 2 (Tschannen et al., 2025), except when producing the high-resolution image (size changed from 512 to 1024). Finally, as aforementioned, we adopt the strict L1 loss to optimize our features. It takes only 34 hours on 32 A100 GPUs with BFloat16 training.

We evaluate on diverse MLLM datasets (Zhang et al., 2024): including DocVQA (Mathew et al., 2021), ChartQA (Masry et al., 2022), TextVQA (Singh et al., 2019), InfoVQA (Mathew et al., 2022), TextCaps (Sidorov et al., 2020), HRBench (Wang et al., 2025), RealWorldQA (x.ai, 2024), GQA (Hudson & Manning, 2019), MME Perception (Fu et al., 2025), POPE (Li et al., 2023), MMBench (Liu et al., 2024c), ScienceQA-IMG (Lu et al., 2022), and AI2D (Kembhavi et al., 2016).

## 4.1 COMPARISON WITH SIGLIP 2

Our framework is built on the highly capable SigLIP 2 model. Through our fine-to-coarse supervision, we aim to further enhance its perception capability in MLLMs. Therefore, we systematically compare our SigLIP-HD encoder with the original SigLIP 2-So400m/16-512px encoder. By default, we adopt the two-stage training pipeline of LLaVA (Liu et al., 2024a). The input image is resized to $512^2$ pixels and encoded into $32^2$ visual tokens to be sent into the LLM.

As compared in Table 5, with Vicuna-1.5-7B (Zheng et al., 2023) as the LLM, we try different supervised fine-tuning (SFT) data, including LLaVA-1.5 (Liu et al., 2024a) and LLaVA-NeXT (Liu et al., 2024b) that contains more OCR-related data. We also attempt different training configurations, including freezing or unfreezing the vision encoder during the fine-tuning stage. Across all settings, our SigLIP-HD consistently outperforms our SigLIP 2 baseline. Notably, on OCR-related benchmarks, such as DocVQA and ChartQA, we improve SigLIP 2 from 56.0 → 59.6 (+3.6) and from 61.6 → 65.2 (+3.6), respectively. On general VQA benchmarks that require fine-grained perception, *e.g.*, HRBench, our SigLIP-HD outperforms its baseline by +4.8 (43.5 → 48.3). Qualitative comparisons are provided in Figure 3. Our SigLIP-HD exhibits better capability in perceiving fine-grained content.

**Using AnyRes.** Although we primarily focus on unleashing the perception capability at a standard (medium) resolution, our encoder is indeed compatible with the AnyRes (Li et al., 2025a) strategy

Table 7: Comparison between our SigLIP-HD and SigLIP 2 under **more LLMs**. Here we use the LLaVA-NeXT data for visual instruction tuning and freeze the vision encoder.

| LLM | Encoder | DocVQA | ChartQA | TextVQA | InfoVQA | TextCaps | GQA | POPE | SQA[I] | Avg |
|-----|---------|--------|---------|---------|---------|----------|-----|------|--------|-----|
| Llama-3.2-3B | SigLIP 2 | 47.3 | 45.2 | 59.2 | 22.2 | 66.1 | 59.2 | 88.3 | 70.4 | 57.2 |
| | SigLIP-HD | **49.9** | **49.8** | **60.8** | **23.9** | **66.3** | **60.2** | **88.5** | **71.4** | **58.9** |
| Qwen2.5-7B | SigLIP 2 | 62.5 | 66.1 | 66.1 | 30.6 | 68.4 | 64.0 | 88.3 | 79.6 | 65.7 |
| | SigLIP-HD | **64.2** | **66.8** | **66.3** | **30.8** | **68.7** | **64.2** | **88.6** | **79.9** | **66.2** |

Table 8: Ablation study on the type of feature alignment loss. We adopt L1 loss for its better results.

| Feature alignment loss | DocVQA | ChartQA | TextVQA | InfoVQA | TextCaps | GQA | POPE | SQA[I] | Avg |
|------------------------|--------|---------|---------|---------|----------|-----|------|--------|-----|
| Cosine similarity (Fang et al., 2023) | 33.8 | 20.2 | 62.5 | **23.8** | 71.3 | **64.4** | 88.0 | 70.1 | 54.3 |
| Cosine sim + smooth L1 (Ranzinger et al., 2024) | 34.1 | **20.9** | 62.9 | 23.2 | 70.6 | 63.8 | 87.8 | 70.5 | 54.2 |
| L1 | **34.7** | 20.2 | **63.1** | 23.0 | **71.6** | 64.3 | **88.1** | **71.5** | **54.6** |

Table 9: Ablation study on the image scale used to supervise our SigLIP-HD encoder. This is different from the experiment in Table 2, as we here study the scales directly under our training framework.

| Image scale | DocVQA | ChartQA | TextVQA | InfoVQA | TextCaps | GQA | POPE | SQA[I] | Avg |
|-------------|--------|---------|---------|---------|----------|-----|------|--------|-----|
| 1 scale ($1024^2$) | 27.8 | 18.6 | 51.7 | 21.4 | 68.9 | 62.2 | 88.0 | 67.3 | 50.7 |
| 2 scales ($512^2+1024^2$) | **34.7** | **20.2** | **63.1** | **23.0** | **71.6** | **64.3** | 88.1 | **71.5** | **54.6** |
| 3 scales ($512^2+1024^2+1536^2$) | 33.4 | 19.4 | 62.7 | 22.9 | 70.8 | 64.0 | **88.2** | 70.6 | 54.0 |

(*i.e.*, operating on native resolution). We report the results in Table 6. We choose the closest resolution from $512 \times \{\{1 \times 1\}, \cdots, \{3 \times 3\}\}$. The max tokens are set as $32^2 \times 4$. While the baseline method has been significantly enhanced by such an AnyRes strategy, our SigLIP-HD can further outperform it by aN even larger margin on OCR tasks, *e.g.*, improving ChartQA from 63.9 to 67.4 (+3.5).

**More LLMs.** In addition to the widely used Vicuna-1.5 (Zheng et al., 2023), which is fine-tuned from Llama 2 (Touvron et al., 2023), we further extend our vision encoder to other LLMs. As shown in Table 7, with a smaller Llama-3.2-3B LLM (Grattafiori et al., 2024), our SigLIP-HD still showcases clear advantages. It surpasses SigLIP 2 by +4.6 ($45.2 \rightarrow 49.8$) on ChartQA, +2.6 ($47.3 \rightarrow 49.9$) on DocVQA, and +1.6 ($59.2 \rightarrow 60.8$) on TextVQA. Beyond the Llama series, with the latest Qwen2.5-7B (Yang et al., 2024) as the LLM, our SigLIP-HD is also consistently superior to our baseline encoder, *e.g.*, boosting it from $62.5 \rightarrow 64.2$ (+1.7) on DocVQA.

## 4.2 ABLATION STUDY

We use the LLaVA-1.5 data (Liu et al., 2024a) for instruction tuning and freeze the vision encoder.

**Feature alignment loss.** Existing works have provided several candidate choices on the loss function used for feature alignment, such as the cosine similarity loss adopted by EVA (Fang et al., 2023) and the cosine similarity + smooth L1 loss adopted by AM-RADIO (Ranzinger et al., 2024). In Table 8, we compare these losses, as well as the plain L1 loss. These three losses deliver very close results on average, all surpassing our SigLIP 2 baseline. Nevertheless, the simplest and strictest L1 loss is slightly better than the other two losses. So we choose it as our final feature alignment loss.

**Image scale.** From our pilot studies (Table 2), we can conclude that feeding multi-scale images of two scales or three scales to the MLLM performs the best. Here, we further examine the final performance of our SigLIP-HD encoder when trained under different configurations of multiple scales. As shown in Table 9, similar to our observations in pilot studies, the two-scale configuration (one base-scale $512^2$px image + one high-resolution $1024^2$px image) achieves the best results, providing the most suitable features for our SigLIP-HD to learn. We believe there may be more sophisticated and better configurations, *e.g.*, searching for the optimal fusion weight for the three-scale or even four-scale features, but they are out of the scope of this paper. We aim to provide a neat, universal, and efficient training framework without too many hand-crafted hyper-parameters.

**Multi-scale fusion weight.** We find the simplest "interpolate + average" practice delivers better results than pixel unshuffle and channel-wise concatenation in our pilot studies (Table 4). Here we

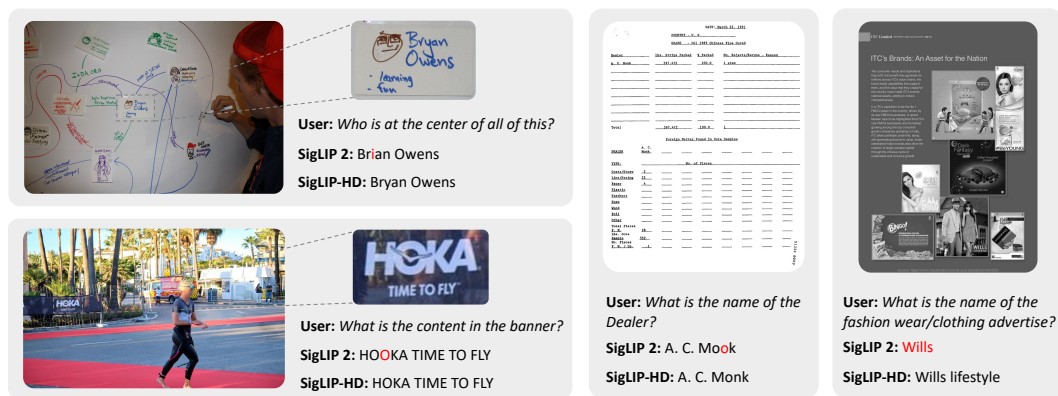

Figure 3: Qualitative comparison between our SigLIP-HD and SigLIP 2 when applied in MLLMs.

Table 10: Ablation study on the fusion weight for averaging base-scale and high-resolution features.

| Base:high | DocVQA | ChartQA | TextVQA | InfoVQA | TextCaps | RWQA | GQA | MME$^P$ | POPE | SQA$^I$ | Avg |
|---|---|---|---|---|---|---|---|---|---|---|---|
| 1:1 | **34.7** | **20.2** | **63.1** | 23.0 | **71.6** | **59.5** | **64.3** | **1554.0** | **88.1** | **71.5** | **57.4** |
| 1:2 | 32.6 | 19.8 | 61.5 | 22.0 | 71.3 | 57.1 | 62.7 | 1513.4 | 88.0 | 68.8 | 55.9 |
| 2:1 | 33.6 | 19.5 | 61.3 | **23.5** | 70.9 | 57.5 | 63.9 | 1544.0 | 87.9 | 69.2 | 56.5 |

Table 11: Shifting our baseline model from SigLIP 2 (Tschannen et al., 2025) to OpenAI-CLIP (Radford et al., 2021). For fair comparison, we feed CLIP with multi-scale inputs and keep the same number of visual tokens as our CLIP-HD.

| Encoder | Resolution | Tokens | DocVQA | ChartQA | TextVQA | InfoVQA | TextCaps | GQA | MME$^P$ | POPE | SQA$^I$ | Avg |
|---|---|---|---|---|---|---|---|---|---|---|---|---|
| CLIP | $336^2$ | $24^2$ | 22.4 | 17.5 | 46.8 | 20.8 | 69.0 | 63.0 | **1520.9** | 86.9 | 70.2 | 52.5 |
| CLIP | $336^2+672^2$ | $37^2$ | 31.1 | 18.1 | 52.0 | **22.3** | 69.5 | 63.0 | 1510.7 | 87.3 | 70.0 | 54.3 |
| CLIP-HD | $518^2$ | $37^2$ | **33.2** | **20.4** | **53.0** | 21.5 | **70.2** | **63.3** | 1519.6 | **87.8** | **70.8** | **55.1** |

further validate whether mean average is better than weighted average. As compared in Table 10, we attempt to increase the weight of high-resolution features or the weight of base-scale features. As a result, neither of them outperforms the default mean average strategy. Slightly to our surprise, even on benchmarks that require fine-grained details, *e.g.*, DocVQA, increasing the weight of high-resolution features still deteriorates the final result. This further proves the indispensable role of base-scale images in capturing the global content.

### 4.3 COMPARISON WITH LEGACY MODEL OPENAI-CLIP

Until now, we have provided comprehensive results to demonstrate the superiority of our SigLIP-HD over SigLIP 2. To be more convincing, we here further validate our fine-to-coarse supervision methodology on the legacy model OpenAI-CLIP-L/14-336px (Radford et al., 2021). However, as illustrated in Figure 1, it is impossible even for humans to recognize detailed image content under the 336px low resolution. Therefore, our model is trained at $518^2$ pixels ($37^2$ visual tokens) by interpolating the pre-trained positional embeddings. The high-quality features used for learning are produced under three image scales ($336^2+672^2+1008^2$). We interpolate the three-scale features to the same shape ($37^2$) and then average them to supervise our CLIP-HD.

To ensure a fair comparison due to our larger resolution and more visual tokens, when using the pre-trained CLIP in MLLMs, we feed a multi-scale input ($336^2+672^2$) following LLaVA-NeXT (Liu et al., 2024b). The obtained two-scale features are both interpolated to the same size as ours and then averaged to be sent into LLM. As compared in Table 11, although multi-scale inputs and increased visual tokens have significantly boosted the results on OCR-related benchmarks, our CLIP-HD can further enhance the performance at the same cost, *e.g.*, 22.4 → 31.1 → 33.2 on DocVQA. Our superior results based on the legacy model CLIP and the latest SOTA model SigLIP 2 demonstrate the universality of our proposed fine-to-coarse supervision mechanism.

## 5 RELATED WORK

**Visual representation learning.** Learning robust visual representation has been a fundamental goal in computer vision for decades. The rise of deep learning revolutionized the field (Lowe, 2004), starting with supervised learning (Krizhevsky et al., 2012). But human labels are inherently biased (Yun et al., 2021), not enough to produce transferable features. Therefore, CLIPs (Radford et al., 2021) leverage web captions to learn more informative representations. However, such alt-text data is often noisy, prompting continued interest in vision-centric self-supervised learning (Caron et al., 2021). This line of work, popularized by contrastive learning (Chen et al., 2020) and masked image modeling (He et al., 2022), remains highly active. Thanks to large-scale curated data (Xu et al., 2024) and hybrid supervision signals (Oquab et al., 2024), latest works (Tschannen et al., 2025; Bolya et al., 2025) demonstrate the potential of a universal representation for diverse downstream tasks.

This work does not aim to propose a new pre-training strategy, but to further enhance the capability of a pre-trained encoder with a simple and efficient framework.

**Multi-modality large language model (MLLM).** There are two mainstream roadmaps to deal with visual inputs: encoder-based and encoder-free. Encoder-based MLLMs (Liu et al., 2024a; Chen et al., 2024b; Li et al., 2025b) use a pre-trained vision encoder (Radford et al., 2021; Zhai et al., 2023) to extract visual tokens for LLMs. They can leverage rich pre-trained visual knowledge. But they are not unified, and the visual inputs are constrained by the pre-trained resolution. To bypass these limitations, encoder-free MLLMs (Bavishi et al., 2023; Diao et al., 2024; 2025; Lei et al., 2025) are attracting growing research interest. Unfortunately, until now, they require more training budget and still lag behind encoder-based models. Therefore, our work on seeking better visual representations for MLLMs is still of high value.

**Visual representation in MLLMs.** There are broadly three approaches to enhancing visual representation in MLLMs. The first involves directly pre-training more powerful vision encoders (Fan et al., 2025; Bolya et al., 2025). Despite the success, their high training costs remain prohibitive for most researchers. The second approach combines multiple pre-trained encoders (Tong et al., 2024b), aiming for mutual gains. In practice, unfortunately, they rarely yield substantial gains over a single well-optimized encoder (Shi et al., 2025b). The third and most prevalent strategy is scaling up image resolution (Chen et al., 2024a; Bai et al., 2025). Early works resize images to a fixed larger size (Liu et al., 2024a), but recent techniques preserve native resolutions (Bai et al., 2025). Beyond global resizing, local zoom-in methods (Zhang et al., 2025; Qian et al., 2025) can also amplify details.

In contrast to this scaling trend, our work takes a step back, demonstrating that even at a standard resolution, fine-grained perception can be achieved effectively.

**Knowledge distillation.** Our work shares core spirit with knowledge distillation (Hinton et al., 2015), in that we both learn from better "teacher" tokens. However, we do not rely on any external knowledge of an auxiliary model. Instead, we unleash the inherent potential of the current model. We eliminate the need to align multiple models into a shared space (Ranzinger et al., 2024; Heinrich et al., 2025). CLIPSelf (Wu et al., 2024a) similarly refines CLIP features. But it uses region-level coarse supervision and targets at the open-vocabulary dense prediction, while we adopt patch-wise fine-grained supervision and address the key challenge of MLLMs. Lastly, our work is slightly related to a recent work (He et al., 2025). But we address fundamentally different problems (MLLM fine-grained perception *vs.* depth estimation) and operate on different spaces (feature *vs.* label).

While knowledge distillation or teacher-student training is well-established techniques, ***what to distill*** is critical for effective learning. Our key insight lies in presenting a multi-resolution to standard-resolution distillation framework that enhances representation quality without increasing inference cost. This is orthogonal to existing multi-teacher distillation works (Ranzinger et al., 2024; Heinrich et al., 2025). We demonstrate that multi-resolution distillation can serve as an effective, lightweight post-training stage for enhancing fine-grained understanding without architectural changes or inference overhead. It can be a practical solution for resource-constrained deployment scenarios.

## 6 CONCLUSION

In this work, we reflect on the scaling trend of image resolution in MLLMs. Motivated by the amazing capability of the human visual system, we investigate how to enhance the perception capability of

AI systems at a standard (medium) resolution, without using larger images. We present a highly simple yet effective fine-to-coarse supervision mechanism to address this. Features of the base-scale image are enforced to mimic the high-quality ensembled features of multi-scale images. Built on the latest SigLIP 2 encoder, our fine-tuned SigLIP-HD encoder delivers stronger results across extensive MLLM benchmarks under various training protocols, especially for OCR-related tasks.

**Acknowledgment.** This work is supported by the National Natural Science Foundation of China (No. 62422606, 62441615) and Hong Kong Research Grant Council General Research Fund (No. 17213925).

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
