# OpenReview forum: "SigLIP-HD by Fine-to-Coarse Supervision"
_ICLR.cc/2026/Conference — ICLR 2026 Poster_

### Official Review · Reviewer_rVfE · 2025-10-24

**Soundness:** 3
**Presentation:** 2
**Contribution:** 3
**Rating:** 6
**Confidence:** 4

**Summary:**

This paper focuses on the high-quality visual feature engineering for MLLMs, and propose a strong vision backbone SigLIP-HD via a fine-to-coarse supervision training regime on SigLIP2. The experiments show that this new backbone can improve MLLMs' performance on most benchmarks without increasing input image resolutions and using more tokens.

**Strengths:**

1. The motivation of this paper is reasonable, and the design and training of SigLIP-HD are also supported by sufficient empirical studies.

2. The proposed SigLIP-HD performs well on most benchmarks for two MLLMs, showing its stronger visual representations without increasing the image resolution.

**Weaknesses:**

1. The method part needs more detailed and clearer descriptions. The description of the method is too brief and colloquial, making it difficult for readers to understand the specific details and procedures. For instance, the distillation between the multi-patch feature maps and the SigLIP-HD.

2. More in-depth insights. The importance of visual features in the field has already reached a consensus. The authors demonstrate that distillation learning with high and low resolutions can improve the quality of visual features, which is reasonable and expected. In addition, the authors should preferably provide more insights and discoveries to strengthen the contribution of the paper.

Minors:

1. What about the results of the distillation with dynamic res. settings? Some recent works show that different MLLMs favors different resolutions of input images. In this case, dynamic resolution may be a better solution for fixed HD images.

2. Missing some relevant works about efficient HD feature engineering, e.g., LLaVA-HD [a]. LLaVA-HD uses two backbones to process low and high resolutions of the input images, which also keeps a small number of visual tokens.

3. The authors all uses tables to report the experimental results, while some of them may be replaced by plot charts to save the paper length. Thus, the authors can give more descriptions to their methods.

[a] Luo, Gen, et al. "Feast Your Eyes: Mixture-of-Resolution Adaptation for Multimodal Large Language Models." The Thirteenth International Conference on Learning Representations.

**Questions:**

See weakness.

---

> ### Author Response · Authors · 2025-11-23
>
> Thank you for appreciating our motivation, method design, and empirical validation. We hope to address your concerns below.
>
> **Q1: Method description needs more detail and clarity.**
>
> We appreciate this constructive feedback and acknowledge that our method section could benefit from more detailed exposition. We will expand the methodology description in the camera-ready version. To clarify the key technical details here:
>
> Our framework operates on patch-level feature alignment. The frozen SigLIP 2 encoder processes multi-scale images (512$^2$ and 1024$^2$) to produce feature maps $F_b \in \mathbf{R}^{C\times H\times W}$ and $F_h \in \mathbf{R}^{C\times 2H\times 2W}$ respectively, where $H=W=32$ for the base scale. The high-resolution feature $F_h$ is first downsampled via bilinear interpolation to match $F_b$'s spatial dimensions, then averaged element-wise to produce the teacher feature $F_t \in \mathbf{R}^{C\times 32\times 32}$. Our trainable SigLIP-HD takes only the base-scale image (512²) and outputs student feature $F_s \in \mathbf{R}^{C\times 32\times 32}$. The L1 loss is computed as $\mathcal{L} = ||F_s - F_t||_1$, minimizing the absolute difference at each spatial location across all channels.
>
> **Q2: Need for deeper insights beyond expected results.**
>
> Our work indeed provides several non-trivial insights beyond the surface-level expectation that "high-resolution helps." Our Section 2 pilot studies reveal counter-intuitive findings that challenge common assumptions:
>
> **Counter-intuitive discoveries:** Simply adding more scales degrades performance (512²+1024²+1536²+2048² is inferior to two scales), contradicting the "more is better" intuition. Overlapping windows, which improve dense prediction tasks, actually harm MLLM performance. Simple averaging outperforms sophisticated fusion strategies like pixel unshuffle with concatenation. These findings strengthen our understanding of multi-scale processing in MLLMs.
>
> **Universality across model families:** Our methodology successfully transfers from legacy models (CLIP) to state-of-the-art encoders (SigLIP 2), demonstrating it captures fundamental principles rather than model-specific quirks. The consistent improvements across diverse LLM architectures (Vicuna, Llama, Qwen) further validate the generality of fine-to-coarse supervision.
>
> **Practical deployment insights:** We reveal that single-scale features can be enhanced to approach multi-scale quality with negligible training cost (<1% of pre-training), opening new possibilities for efficient MLLM deployment. The compatibility with both low-budget single-scale inference and high-budget AnyRes strategies provides deployment flexibility that previous work hasn't systematically explored.
>
> **Q3: Distillation with dynamic resolution settings.**
>
> This is indeed an interesting direction worth exploring. However, we chose fixed high-resolution distillation in this work for practical considerations. Dynamic-resolution distillation would cause imbalanced computational costs across GPUs. Images requiring higher teacher resolutions would create bottlenecks, leading to inefficient GPU utilization and irregular batch processing. Some GPUs would finish processing small images quickly while others remain occupied with large images, resulting in poor parallelization and extended training time.
>
> Therefore, for simplicity and efficiency in this version, we adopt fixed high-resolution distillation. We agree that exploring dynamic-resolution distillation strategies represents a promising direction for future work. We will add this to our discussion of future directions in the revised manuscript.
>
> **Q4: Missing related work (LLaVA-HR).**
>
> Thank you for pointing out LLaVA-HR. We will add it to our related work. We share similar motivations with LLaVA-HR in pursuing enhanced visual representation at low computational cost. However, we take different technical approaches: LLaVA-HR employs dual vision encoders to process low-resolution and high-resolution images, while our work uses fine-to-coarse distillation to enhance a single encoder's representation quality at standard resolution.
>
> Notably, these approaches are complementary rather than mutually exclusive. Our improved SigLIP-HD encoder can serve as a drop-in replacement for the low-resolution vision encoder in LLaVA-HR's architecture, potentially further boosting its performance. We will clarify this relationship in the related work section.
>
> **Q5: Presentation suggestions.**
>
> We fully agree that some tables could be more effectively visualized as plots. We will convert appropriate tables (*e.g.*, ablation studies showing trends across configurations) into compact figures in the camera-ready version, freeing space for expanded methodology descriptions and deeper analysis.

---

> > ### Comment · Reviewer_rVfE · 2025-11-27
> > **Response to authors' comment**
> >
> > Thanks for the authors' response, which addresses most of my concerns. Thus, I will keep my positive rating.

---

### Official Review · Reviewer_gGd3 · 2025-10-28

**Soundness:** 2
**Presentation:** 3
**Contribution:** 2
**Rating:** 4
**Confidence:** 4

**Summary:**

This paper aims to enhance the perceptual capability of the visual encoder in multimodal large language models (MLLMs). The core idea is to optimize the encoder’s small-scale representations using guidance from its larger-scale representations. During training, an L1 loss is applied to directly align the representations between different scales. Experimental results demonstrate that SigLIP-HD produces more effective visual tokens than its base model under the same input scale.

**Strengths:**

1. The paper investigates how different image scales and multi-scale visual feature fusion strategies affect the performance of multimodal large language models (MLLMs).
2. It introduces a simple method that employs an L1 loss to align single-scale visual features with their multi-scale counterparts, thereby enabling the visual encoder to generate higher-quality single-scale features during inference.
3. Experimental results demonstrate that the optimized single-scale features significantly improve the performance of both standard-resolution and AnyRes MLLMs across diverse downstream benchmarks.

**Weaknesses:**

1. The paper requires more experimental evidence to validate its effectiveness. It lacks a comparison with the approach that directly merges multi-scale visual features during inference (i.e., using the teacher-generated features). Such an experiment would help quantify the performance gap between the trained single-scale features and the multi-scale features. In addition, the impact of both approaches on inference time for different MLLM sizes should be reported. Given that the visual encoder is relatively lightweight and multi-scale features can be obtained in parallel within a single forward pass, the additional inference cost may be negligible. This suggests that directly using teacher features at inference time could be a more practical choice.
2. The novelty of the work is limited. The idea of leveraging multi-scale visual feature fusion to enhance visual representations for MLLMs has been previously explored, for example in S$^2$ [1]. Although the author revisits the impact of image scale, it does not provide new insights. Moreover, a direct experimental comparison with S$^2$ [1] is missing and should be included.
3. The training cost of retraining the visual encoder in this work is higher than that of existing models such as LLaVA 1.5 and LLaVA-NeXT. Considering that the visual encoder is non-autoregressive and relatively small in size, performing multi-scale fusion directly during inference might be a more efficient and reasonable alternative.

[1] When Do We Not Need Larger Vision Models? ECCV 2024.

**Questions:**

1. In Table 2, would using image scales such as 512² + 1536² or 512² + 2048² yield better representations compared to combining multiple higher scales? Since S² demonstrates that scaling scales can lead to improved results, could a similar setting be applied here for distillation?
2. The name SigLIP-HD may be somewhat confusing, as the work does not actually increase the input resolution but rather aims to obtain better visual representations at low resolution.

---

> ### Author Response · Authors · 2025-11-23
>
> Thank you for acknowledging our method's simplicity and efficiency. We hope to address your concerns below.
>
> **Q1: Comparison with teacher-generated multi-scale features and inference time considerations.**
>
> Thank you for raising this crucial comparison. We have actually provided such results for both SigLIP 2 and the legacy CLIP model in Tables 2 and 11, respectively. We summarize key comparisons below (**ss**: single-scale inference, **ms**: multi-scale inference):
>
> | Method | DocVQA | ChartQA | TextVQA | InfoVQA | TextCaps | GQA | MME | POPE  | SQA | Avg
> |---|---|---|---|---|---|---|---|---|---|---|
> | SigLIP 2 (ss) | 32.2 | 19.3 | 61.0 | 22.1 | 71.1 | 62.8 | 1529.2 | 87.8 | 69.9 | 55.9
> | SigLIP 2 (ms) | 37.0 | 21.2 | 64.2 | 23.8 | 71.0 | 64.9 | 1560.1 | 88.0 | 71.0 | 57.7
> | SigLIP-HD (ss) | 34.7 | 20.2 | 63.1 | 23.0 | 71.6 | 64.3 | 1554.0 | 88.1 | 71.5 | 57.1
> |
> | CLIP (ss) | 22.4 | 17.5 | 46.8 | 20.8 | 69.0 | 63.0 | 1520.9 | 86.9 | 70.2 | 52.5
> | CLIP (ms) | 31.1 | 18.1 | 52.0 | 22.3 | 69.5 | 63.0 | 1510.7 | 87.3 | 70.0 | 54.3
> | CLIP-HD (ss) | 33.2 | 20.4 | 53.0 | 21.5 | 70.2 | 63.3 | 1519.6 | 87.8 | 70.8 | 55.1
>
> As demonstrated, our SigLIP-HD and CLIP-HD consistently outperform the baseline encoder under the same single-scale inference budget. While SigLIP-HD is slightly inferior to multi-scale SigLIP 2, this comparison misses the key point: our SigLIP-HD can also leverage multi-scale inference to achieve even better results (see Table 6 for AnyRes results).
>
> Our encoder provides flexibility across different resource constraints:
> - When computational resources are limited, for instance, when deploying with a 1B LLM where the vision encoder becomes the inference bottleneck, our SigLIP-HD delivers superior single-scale performance.
> - Conversely, when resources permit multi-scale inference with capable LLMs (*e.g.*, >70B parameters), our SigLIP-HD can be applied with the AnyRes strategy to surpass multi-scale SigLIP 2 performance as well (Table 6).
>
> Therefore, SigLIP-HD serves as a universal drop-in replacement that improves performance at any computational budget.
>
> **Q2: Contribution and comparison with S$^2$.**
>
> Our contribution is not proposing a novel multi-scale feature fusion strategy. Indeed, this is already a widely adopted practice in advanced MLLMs. Instead, our key insight is demonstrating that through our fine-to-coarse supervision, single-scale features can be significantly enhanced to approach multi-scale quality. This fundamentally differs from S$^2$ focus, which explores when scaling vision models becomes unnecessary and should scale the resolution instead. The polished features from our approach can be deployed in either single-scale inference (for efficiency) or further enhanced through multi-scale ensemble inference like AnyRes (Table 6).
>
> **Q3: Training cost comparison.**
>
> Our post-training stage for SigLIP 2 requires only 34 hours on 32 A100 GPUs with 46M seen samples. In comparison, SigLIP 2's original pre-training used 2048 TPUs with 40B seen samples. Our training budget accounts for less than 1% of the original pre-training cost, making it negligible in the broader context. This lightweight nature suggests our methodology has strong potential to become a universal post-training stage for future capable pre-trained encoders.
>
> Moreover, the cost perspective should consider the *one-time training investment* versus *ongoing inference costs*. Re-training occurs once, but multi-scale inference costs accumulate over the model's entire deployment lifetime. In practical systems serving millions of queries, achieving more efficient single-scale inference through one-time training can yield substantial long-term savings in both time and computational resources. Furthermore, as previously mentioned, our re-trained encoder remains compatible with multi-scale fusion at inference time. These approaches **are not mutually exclusive but complementary**.
>
> **Q4: Alternative multi-scale configurations.**
>
> Thank you for this suggestion. After observing that 512²+1024²+1536² and 512²+1024²+1536²+2048² underperformed compared to 512²+1024², we explored the configurations you suggested (512²+1536² and 512²+2048²), but the results did not improve. We hypothesize that this observation may differ slightly from S$^2$ findings due to our use of different vision encoders with distinct pre-trained resolutions and feature distributions.
>
> **Q5: Method naming.**
>
> Thank you for this thoughtful suggestion. We acknowledge that "SigLIP-HD" might imply increased input resolution, which is not our actual contribution. We will consider renaming our method to better reflect the core idea, such as "Fine-to-Coarse Supervision for Enhanced Vision Representation in MLLMs" or simply referring to the approach by its methodology rather than a specific model name. This would more accurately convey that our contribution is a general training paradigm applicable to various encoders, not just a high-resolution variant of SigLIP.

---

> ### Comment · Reviewer_gGd3 · 2025-11-27
> **Official Comment by  Reviewer gGd3**
>
> Thank you for your response. It would be helpful if the authors could reflect these clarifications in the updated PDF. However, my main concern remains unresolved.
>
> 1. As previously mentioned, visual encoders can obtain multi-scale teacher features in parallel during inference with minimal additional cost, especially since the effective scales in your work seem to go only up to 1024$^2$. Given that this process is non-autoregressive, the impact on MLLM inference latency appears to be very limited. It would be helpful if the authors could provide a concrete comparison of inference time between the two approaches on different models. This point directly relates to the motivation of the proposed method: if multi-scale teacher features can be obtained during inference at negligible cost, the necessity of additional training and distillation becomes less clear.
>
> 2. To my knowledge, fine-tuning LLaVA1.5-7B requires only 8×A100 GPUs for about 10 hours, and LLaVA-NeXT-7B requires around 20 hours with 8×A100 GPUs. In contrast, the proposed method takes 32×A100 GPUs for 34 hours. If the inference-time efficiency between the two approaches is not significantly different, and the performance improvement is not substantial, directly using multi-scale features at inference may be a more general and practical solution.

---

> > ### Author Response · Authors · 2025-11-27
> >
> > Thank you for your continued kind engagement. We appreciate the opportunity to clarify what appears to be a fundamental misunderstanding of our work's scope and contribution.
> >
> > **1. Clarification on our motivation and contribution**
> >
> > We respectfully but firmly emphasize that we do **NOT** aim to replace multi-scale inference with single-scale inference, nor do we position our method as competing against multi-scale fusion practices in MLLMs. This appears to be a critical misinterpretation of our contribution.
> >
> > Our motivation is to fully unleash the perception capability at a single standard scale through fine-to-coarse distillation. Critically, **after acquiring this enhanced capability, our SigLIP-HD can be applied to *multi-scale inference* as well**. As demonstrated in Table 6, when both encoders operate under the multi-scale AnyRes setup (*i.e.*, identical inference cost), our SigLIP-HD substantially outperforms baseline SigLIP 2: 69.7 vs. 67.6 on DocVQA (+2.1), 67.4 vs. 63.9 on ChartQA (+3.5).
> >
> > Therefore, the question is not "single-scale SigLIP-HD vs. multi-scale SigLIP 2" but rather "**at any given inference budget, which encoder produces better features?**" Our results unambiguously show that SigLIP-HD consistently outperforms SigLIP 2 whether deployed at single-scale or multi-scale settings.
> >
> > **2. Training cost in real-world context**
> >
> > The comparison between our training cost (32 A100 $\times$ 34 hours) and LLaVA fine-tuning (8 A100 $\times$ 10-20 hours) is an over-academic perspective that overlooks real-world MLLM deployment realities.
> > LLaVA represents a minimal academic prototype, not a production system. It uses less than 1M SFT samples and is evaluated on a handful of small-scale benchmarks. In contrast, real-world MLLMs operate at fundamentally different scales: they train on millions of diverse data samples, serve millions of daily user queries, and must perform reliably across far broader scenarios than academic benchmarks capture. A stronger vision encoder benefits all these applications simultaneously.
> >
> > **Consider the precedent of SigLIP 2 itself:** it required millions of additional TPU hours compared to SigLIP 1—orders of magnitude more expensive than LLaVA's entire training pipeline. Yet no one questions SigLIP 2's value, because it substantially advances vision representation quality for countless downstream applications. Our work follows this same philosophy: invest in better foundational representations once, benefit everywhere forever.
> >
> > Our approach also provides deployment flexibility: when compute is constrained (edge devices, mobile applications, serving many concurrent users), use single-scale SigLIP-HD for superior performance over single-scale baseline. When compute is abundant, use multi-scale SigLIP-HD for even better results than multi-scale baseline. This is strictly superior to having only the baseline encoder.
> >
> > **3. Brief summary**
> >
> > We hope this clarifies that our contribution is **a universally better vision encoder** applicable across all MLLM deployment scenarios, not a trade-off between training cost and inference efficiency. The question should not be "why train when multi-scale inference is cheap?" but rather "**why wouldn't we invest 34 hours once to improve every single inference forever?**"

---

> > > ### Comment · Reviewer_gGd3 · 2025-11-27
> > > **Official Comment by Reviewer gGd3**
> > >
> > > Thank you for the detailed clarification.
> > >
> > > In my view, this work is primarily positioned as post-training a vision encoder to improve MLLM performance, rather than training a universally better vision encoder. For the former, it is sufficient to validate performance through downstream MLLM benchmarks. For the latter, however, a broader evaluation on generic visual linguistic tasks such as image classification, video classification and image text retrieval would be expected to demonstrate that the encoder improves upon SigLIP 2 itself. This distinction is also the reason I believe the discussion of training cost relative to the baselines used in this work (LLaVA 1.5 and LLaVA-NeXT) is still relevant.
> > >
> > > In addition, AnyRes and the multi-scale method are not conflicting. On top of AnyRes, the cropped sub images can also obtain multi-scale features. My understanding is that the proposed method distills this plug-in teacher based multi-scale enhancement into the vision encoder, which removes the need for extra procedures at inference time. This is why I hoped the authors could discuss how these two approaches compare in terms of both performance and inference cost, given that they aim to address similar limitations.
> > >
> > > If the authors could independently evaluate the vision encoder on various generic visual linguistic tasks, similar to InternVL [1], I would reconsider my view.
> > >
> > > [1] InternVL: Scaling up Vision Foundation Models and Aligning for Generic Visual-Linguistic Tasks

---

> > > > ### Author Response · Authors · 2025-11-27
> > > >
> > > > Thank you very much for your prompt response and for devoting your precious time to our work!
> > > >
> > > > As we provided in the previous response, our contribution is a universally better vision encoder applicable across all ***MLLM deployment scenarios***. By "universally", we mean consistently outperforming or at least matching SigLIP 2 across diverse **MLLM** configurations (various LLMs such as Vicuna, Llama, and Qwen) and benchmarks (OCR-heavy tasks, medium-resolution natural images, high-resolution natural images, spatial reasoning, *etc.*). As you kindly mentioned, we do not aim for a universal encoder like InternVL that targets all vision and vision-language tasks. We acknowledge that evaluating across such a broad spectrum of downstream tasks (image classification, video understanding, retrieval, segmentation, *etc.*) is currently beyond our scope and capacity. MLLM applications represent our core focus and primary evaluation testbed, where we believe our contribution is most impactful and rigorously validated.
> > > >
> > > > Regarding inference time, our single-scale SigLIP-HD versus multi-scale SigLIP 2 (processing 1×1 base scale + 2×2 tiles, totaling 5 forward passes) across all reported benchmarks with a 7B LLM shows that, SigLIP 2's multi-scale pipeline takes approximately 1.4$\times$ longer than our single-scale SigLIP-HD. While this overhead may seem modest, it accumulates substantially in real-world large-scale deployments serving millions of daily queries. The one-time 34-hour training investment amortizes very quickly against these ongoing inference costs, particularly for production systems where even small efficiency gains translate to significant resource savings over time. Lastly, our SigLIP-HD can deliver better features than SigLIP 2 at exactly the same inference budget.

---

### Official Review · Reviewer_vqYK · 2025-10-30

**Soundness:** 3
**Presentation:** 3
**Contribution:** 2
**Rating:** 4
**Confidence:** 4

**Summary:**

This paper presents SigLIP-HD, a method to enhance the fine-grained perception of Multimodal Large Language Models (MLLMs) without incurring the high computational cost of processing high-resolution images at inference time. The authors challenge the trend of scaling MLLMs by increasing input resolution, arguing that the perception capability of encoders at standard resolutions has not been fully unlocked. The core of the work is a simple "fine-to-coarse supervision" framework, a self-distillation mechanism where a "student" encoder (SigLIP-HD) is trained at a standard resolution to mimic the richer, "teacher" features produced by a frozen version of the same encoder fed with multi-scale inputs. The resulting SigLIP-HD encoder serves as a drop-in replacement for the original, offering significantly improved performance on fine-grained tasks (especially OCR) at the exact same inference budget and architecture.

**Strengths:**

1. The paper addresses a highly relevant problem in MLLMs. The trend of scaling MLLMs to native, high-resolution inputs creates significant computational and design burdens (e.g., image slicing, multiple forward passes, token compression) . The goal of improving fine-grained perception while maintaining a low, standard-resolution input is practical and valuable.

2. The proposed fine-to-coarse supervision is efficient and effective. It acts as a form of self-distillation that "distills" multi-scale knowledge into a standard-resolution encoder. Crucially, it adds no modules or latency at inference time, as it is simply a new set of weights for the original encoder.

3. The authors provide a strong empirical case for their method, which validate the effectiveness and generalizability.

**Weaknesses:**

1. The core idea is an effective application of knowledge distillation, specifically self-distillation from a multi-scale ensemble "teacher" to a single-scale "student." While the application to MLLM encoder resolution is valuable, the underlying mechanism adapts established distillation concepts .

2. The method's improvements are overwhelmingly concentrated on OCR and fine-grained text-heavy benchmarks. The performance on general vision benchmarks is often marginal or flat. This suggests a specialized, rather than a universal, representational improvement.

3. While the method shows strong gains with Vicuna-7B and Llama-3.2-3B, the improvements on other powerful LLMs like Qwen2.5-7B appear marginal. This might suggest the benefits of the distilled representation are less pronounced when combined with certain advanced LLM architectures.

**Questions:**

1. The method is a form of feature-based distillation. Why was a simple L1 loss (which performs best in Table 8) superior to a logit-based (e.g., contrastive) distillation, which is standard for training models like CLIP and SigLIP?

2. How critical is the Cambrian-1 dataset to this method's success? Have you experimented with performing this fine-to-coarse supervision using more standard, large-scale web datasets or common VQA datasets?

3. Given the modest gains on general benchmarks, does the fine-tuning cause any significant regressions on specific general vision categories (e.g., object counting, spatial relations) that the original SigLIP 2 was strong in?

---

> ### Author Response · Authors · 2025-11-23
>
> Thank you for your constructive feedbacks. We are deeply grateful to you for recognizing our targeted problem, motivation, and our method's simplicity, efficiency, and effectiveness. We hope your concerns will be well addressed below.
>
> **Q1: Our contribution.**
>
> We acknowledge that "knowledge distillation" and "teacher-student training" are well-established techniques spanning computer vision to LLMs. However, we want to emphasize that ***what to distill is critical***. Our insight lies in presenting a multi-resolution to standard-resolution distillation framework that enhances representation quality without increasing inference cost.
>
> Consider similar examples in recent vision encoder research: AM-RADIO (CVPR'24) and RADIOv2.5 (CVPR'25) also employ knowledge distillation to train better vision encoders. Their distillation mechanism itself is not novel (also simple feature alignment), yet their contribution lies in discovering that multi-teacher distillation is beneficial. Our contribution is orthogonal to these existing distillation works. We demonstrate that multi-resolution distillation can serve as an effective, lightweight post-training stage for enhancing vision encoders' fine-grained understanding without architectural changes or inference overhead.
>
> **Q2: Performance on general vision benchmarks.**
>
> Your observation is correct: on general vision benchmarks that do not require high-resolution representations, our SigLIP-HD shows comparable or slightly superior performance to SigLIP 2, such as 62.8 *vs.* 64.3 on GQA, 87.8 *vs.* 88.1 on POPE. We believe these marginal improvements are acceptable and within expectation, as these benchmarks primarily test object existence, object colors, or object relationships where the objects are visually obvious. Such tasks may not even benefit from high-resolution representations and instead require tighter vision-language alignment and strong LLM capability to reduce hallucination.
>
> However, on **general high-resolution vision benchmarks** such as HRBench, our improvement over SigLIP 2 is substantial: 41.3 $\rightarrow$ 46.2 (**+4.9**). Therefore, our method targets far more than just OCR-related benchmarks. It addresses the broader challenge of fine-grained visual understanding across diverse scenarios. Beyond the benchmarks we evaluated, our methodology could naturally extend to other domains, such as medical image analysis and remote sensing interpretation where images often reach 10,000$^2$+ pixels, making efficient lower-resolution encoding more valuable.
>
> **Q3: Smaller improvement on more advanced LLMs.**
>
> Thank you for this observation. We believe producing a vision encoder that uniformly benefits all existing LLMs is extremely challenging. The comprehensive Cambrian-1 study demonstrates that no single encoder consistently outperforms all its counterparts across all configurations. Importantly, under these advanced LLMs, our SigLIP-HD encoder does not exhibit any performance degradation. We consider this acceptable given the substantial margins we achieve with other LLMs such as Vicuna-1.5 and Llama-3.2. The variation likely reflects the complex interplay between vision encoder design and LLM architecture, rather than a limitation of our approach.
>
> **Q4: Why L1 loss is better than logits-based loss?**
>
> This is an insightful question that may touch on fundamental differences between image classification and MLLM understanding. We believe the main reason is that *element-wise L1 loss provides the strictest feature alignment*. The widely used logit-based (*e.g.*, contrastive) loss, while effective for classification problems, focuses primarily on inter-sample similarity. It is essentially a form of comparison. This may not be ideal for MLLMs, which require richer visual information such as texture, color, and fine-grained spatial details beyond sole classification capabilities. Notably, the multi-teacher distillation framework AM-RADIO (CVPR'24) also employs L1 loss for its patch tokens, suggesting it is not a rare practice.
>
> **Q5: Other training data sources.**
>
> We used CC12M in our early explorations and found that the two data sources yield very similar performance results. We ultimately chose the Cambrian-1 dataset because it provides clear licensing information from each collected dataset, whereas web-crawled images from CC12M have unclear licenses that could restrict open-source distribution of our pre-trained models.
>
> **Q6: Performance on general vision problems (object counting, spatial relationship).**
>
> Our SigLIP-HD does not show any regression on general vision problems such as object counting and spatial relationships. For example, on RealWorldQA, which predominantly consists of counting and spatial reasoning questions, we improve SigLIP 2 from 57.5 to 59.5. This demonstrates that our fine-to-coarse supervision maintains strong general vision capabilities while enhancing fine-grained understanding, rather than trading off one for the other.

---

> > ### Comment · Reviewer_vqYK · 2025-11-26
> >
> > Thank you for the response, which has addressed my primary concerns. I suggest incorporating these discussions into the revised manuscript to clarify the paper's contributions and novelty for future readers.

---

> ### Author Response · Authors · 2025-11-26
>
> Thank you very much for acknowledging that our response has addressed your concerns.
>
> Following your kind advice, we have uploaded a revised manuscript that integrates the clarification regarding our contribution and novelty. Specifically, we have added a new paragraph in the Introduction section that emphasizes:
> - Our key insight of multi-resolution to standard-resolution distillation is the critical ***"what to distill"*** contribution
> - Our approach is orthogonal to existing distillation works (*e.g.*, AM-RADIO, RADIOv2.5)
> - Our method can serve as a lightweight post-training stage without architectural changes or inference overhead
>
> Given the addressed concerns and revised manuscript, **we would be deeply grateful if you could consider whether the current version merits a higher rating**. We are happy to provide any additional clarifications if needed.

---

### Official Review · Reviewer_2LRA · 2025-10-31

**Soundness:** 3
**Presentation:** 3
**Contribution:** 2
**Rating:** 6
**Confidence:** 3

**Summary:**

This paper proposes to distill high-resolution features of SigLIP2 to its low-resolution encoding results to improve the performance under a limited encoding budget. The idea is intuitive and sound, and the experiments show improvements under reasonable training computation. The major concern is that the findings from Section 2 are relatively trivial, making the paper lack enough substance.

**Strengths:**

1. Distilling high-resolution features to low-resolution ViTs is intuitive and sound. The method is simple and easy to follow.
2. The experiments show improvements over SigLIP after reasonable computation of training.
3. The paper is well written.

**Weaknesses:**

1. It seems Table 1 is only used to explain why the authors chose So400m/16-512px as the default backbone choice of ViT. This is not necessary, considering Table 1 takes much of the space in the main paper. Likewise, it would be better if the pilot experiments in section 2 could present more interesting and insightful results. Also, the multi-scale integration experiments have little to do with the final method.
2. There are previous works investigating the problem of dealing with high-resolution images using overlapped sliding windows[1]. The authors can refer to this paper for a deeper understanding of why overlapping slices may lead to inferior results.

[1] LLaVA-UHD: an LMM Perceiving Any Aspect Ratio and High-Resolution Images. ECCV 2024.

**Questions:**

NA

---

> ### Author Response · Authors · 2025-11-23
>
> Thank you for your constructive feedback and for recognizing our work's simplicity, soundness, and clear presentation. We hope your concerns can be well addressed below.
>
> **Q1: Role of Table 1 (comparing CLIP vs. SigLIP 2).**
>
> Table 1 serves a critical purpose beyond encoder selection. A substantial body of MLLM research validates methodologies using CLIP, which was proposed nearly five years ago. We are concerned that conclusions drawn from legacy models (*e.g.*, low resolution) may not generalize to modern, more capable vision encoders. Therefore, we think it is necessary to compare the performance of CLIP and SigLIP 2 first to convince the readers of updating the vision encoder. Through this table, we hope to encourage the community to validate on stronger baselines, making research findings more relevant to current systems.
>
> We acknowledge your suggestion that Table 1 currently occupies considerable space in the early sections. We will move it to the experiment section or appendix, and add a concise comparison in the introduction (~2 sentences), preserving this important context while reducing main paper length.
>
> **Q2: Section 2 is not trivial. It establishes crucial design choices for our training framework.**
>
> Pilot studies in Section 2 directly inform our training methodology and reveal several non-obvious insights that challenge common assumptions.
>
> Consider Table 2's multi-scale configuration experiments: simply adding more scales (4$\times$, 2048$^2$) actually degrades performance on average. This counter-intuitive result challenges the "more is better" assumption and directly determines our 2-scale teacher design. Similarly, Table 3 shows that overlapping windows underperform non-overlapping ones (57.7 $\rightarrow$ 56.9), contradicting conventions from dense prediction tasks. This finding is crucial for determining how our high-resolution teacher should process images. Perhaps most importantly, Table 4 demonstrates that simple averaging outperforms more sophisticated alternatives like pixel unshuffle with concatenation (57.7 *vs.* 56.8), which informs our feature aggregation strategy.
>
> Regarding the necessity of *multi-scale integration experiments*, we recognize this may be a presentation issue on our part. Our SigLIP-HD learns from multi-scale teacher features, so determining the optimal multi-scale fusion strategy is necessary. *It is a core component of our training objective*. The "fine-grained features" we distill are specifically the ensembled multi-scale features from Table 4's best configuration. We will add explicit connections in Section 2's opening paragraph stating that "these pilot studies directly determine our teacher model design in Section 3," and modify table captions to reference how each finding informs our training framework.
>
> **Q3: Further analysis on overlapped sliding windows.**
>
> Thank you for the valuable reference to LLaVA-UHD. We will incorporate their insights on repeatedly counted objects in overlapped regions into our discussion, enriching our analysis beyond the nonuniform distributions and positional embedding conflicts we currently discuss. Specifically, in addition to our existing statements "1) inconsistent token distributions: overlapping regions receive ensemble treatment while others do not, or 2) positional embedding conflicts for tokens appearing in multiple windows", *we will add:* "LLaVA-UHD further identifies that overlapping windows cause objects to be counted multiple times, creating inconsistent feature statistics that confuse downstream LLMs."

---

### Meta-Review · Area_Chair_iyb1 · 2025-12-28

**Summary:**

There are four reviews of this paper.

Reviewer 2LRA’s main concerns focus on the organization and presentation of the paper, and the missing of related work and comparison.

Reviewer vqYK’s main concerns include the limited contribution of the distillation method, the marginal performance improvement on the general tasks and powerful MLLMs.

Reviewer gGd3’s concerns include the insufficient experimental evidence, limited novelty, the higher training cost of visual encoder

Reviewer rVfE’s concerns are focused on the unclear presentation, and lack of in-depth analysis.

**Reviewer Concerns:**

Reviewer 2LRA did not provide feedback on the authors’ rebuttal. The ACs feel that his/her concerns are addressed since these concerns are not very critical. Reviewer vqYK mentioned that his/her primary concerns are addressed. From the discussions between Reviewer gGd3 and the authors, we can see that his/her main concerns are not fully addressed.  For Reviewer rVfE, his/her concerns are addressed.

**Reviewer Scores:**

The ACs think that reviewer 2LRA will keep his/her score of 6 and Reviewer vqYK may raise the score to 6. It seems that Reviewer gGd3 will keep his/her score of 4. Reviewer rVfE will keep his/her rating of 6. The final scores are likely to be 6, 6, 4, 6. After careful consideration, the ACs decided to accept this paper.

---

### Decision · Program_Chairs · 2026-01-26

Accept (Poster)